# Assessing Human Post-Editing Efforts to Compare the Performance of Three Machine Translation Engines for English to Russian Translation of Cochrane Plain Language Health Information: Results of a Randomised Comparison

**Liliya Eugenevna Ziganshina** [1,2,3,4,*] , **Ekaterina V. Yudina** [1,5], **Azat I. Gabdrakhmanov** [1,3] **and Juliane Ried** [6]

1 Cochrane Russia, St Albans House, 57-59 Haymarket, London SW1Y 4QX, UK; cochranerussia@gmail.com (E.V.Y.); cat.iquality@gmail.com (A.I.G.)
2 Centre for Knowledge Translation, Institute for Methodology of Professional Development, Federal State Budgetary Educational Institution of Continuing Professional Education "Russian Medical Academy of Continuing Professional Education" of the Ministry of Health of the Russian Federation (RMANPO): 2/1, Barrikadnaya Street, 123995 Moscow, Russia
3 Department of Pharmacology, Kazan State Medical University of the Ministry of Health of the Russian Federation (KSMU), 49 Butlerov Street, 420012 Kazan, Russia
4 Department of Basic and Clinical Pharmacology, Kazan Federal University of the Ministry of Science and Higher Education of the Russian Federation (KFU), 18 Kremlevskaya Street, 420008 Kazan, Russia
5 Children's Hospital N 1 of the City of Kazan, 125a Dekabristov Street, 420034 Kazan, Russia
6 Cochrane, 79110 Freiburg, Germany; juliane.ried@cochrane.org
* Correspondence: lezign@gmail.com; Tel.: +7-987-296-8496

**Abstract:** Cochrane produces independent research to improve healthcare decisions. It translates its research summaries into different languages to enable wider access, relying largely on volunteers. Machine translation (MT) could facilitate efficiency in Cochrane's low-resource environment. We compared three off-the-shelf machine translation engines (MTEs)—DeepL, Google Translate and Microsoft Translator—for Russian translations of Cochrane plain language summaries (PLSs) by assessing the quantitative human post-editing effort within an established translation workflow and quality assurance process. 30 PLSs each were pre-translated with one of the three MTEs. Ten volunteer translators post-edited nine randomly assigned PLSs each—three per MTE—in their usual translation system, Memsource. Two editors performed a second editing step. Memsource's Machine Translation Quality Estimation (MTQE) feature provided an artificial intelligence (AI)-powered estimate of how much editing would be required for each PLS, and the analysis feature calculated the amount of human editing after each editing step. Google Translate performed the best with highest average quality estimates for its initial MT output, and the lowest amount of human post-editing. DeepL performed slightly worse, and Microsoft Translator worst. Future developments in MT research and the associated industry may change our results.

**Keywords:** Cochrane Russia; language translation; Russian language; machine translation; machine translation quality; post-editing; volunteer translation; health domain; Cochrane plain language summaries; Google Translate; DeepL; Microsoft Translator

## 1. Introduction

### 1.1. Cochrane and Its Multilanguage Activities

Cochrane is a global, independent, not-for-profit network that collects, assesses, and summarizes health research and publishes the results of its research syntheses, so called Cochrane systematic reviews. Cochrane has published more than 8000 systematic reviews to date, and updates them regularly as new research becomes available. Cochrane reviews aim to help people make informed choices about their health and to improve healthcare globally. Since its foundation in 1993, Cochrane has made a vital contribution to health and

healthcare systems worldwide, and to the development of evidence-based medicine and research synthesis methodology. The work of Cochrane has been unequivocally recognized internationally for informing healthcare decision-making with its high-quality, independent and credible systematic reviews [1,2].

Cochrane reviews are produced in English, and until the early 2010s, the only large-scale, systematic translation activities were into Spanish. This posed a significant barrier to their uptake and use by audiences speaking other languages. In 2013, Cochrane started developing a translation strategy to enable wider access to its health evidence in different languages [3]. Translation and multilanguage activities have become a strategic priority; Cochrane now translates and publishes the scientific abstracts and plain language summaries (PLSs) of its systematic reviews from English into up to 15 languages on a regular basis. It has published more than 30,000 translations as of October 2020. Local teams in different countries manage the translators, workflows and quality assurance for their respective languages. Cochrane publishes about 100 new or updated systematic reviews per month, and each team selects topics most relevant to their local context as capacity allows. Most Cochrane translation teams have very limited or no dedicated funding for their translation activities and rely substantially on volunteer translators and editors. Most volunteers have a background in health, but not in translation. The quality assurance and editing of volunteer translations is a significant burden and often a bottleneck for the teams managing translations.

Cochrane Russia started translating PLSs of Cochrane reviews into Russian in May 2014 [4]. As of December 2020, over 200 volunteers have contributed to Russian translation activities, and the team has published 2724 Russian PLS translations. The Russian translations were viewed more than 1 million times per month from January to September 2020, which demonstrates the demand for Cochrane evidence in Russian.

Cochrane subscribes to the cloud-based translation management system Memsource (https://memsource.com (accessed on 30 December 2020)) to facilitate its translation work. Within Memsource, Cochrane translation teams have access to integrated machine translation engines (MTEs). High-quality machine translation (MT) can facilitate more efficient translation. It is important to understand which MTE performs best for specific languages—and Cochrane content—to make the most of this technology and reduce the editing burden.

*1.2. Machine Translation and Health Information*

Over the last two decades, the potential of MT and its available options have been rapidly growing, with impressive improvements in terms of quality [5,6]. However, the general understanding is that MT is not perfect, and its quality may decrease in some specific medical domains, unless it is domain-adapted. With the rapid growth of telemedicine, the real-life implementation of high-quality MT is anticipated in everyday general medical practice [7].

A systematic review, which aimed at characterizing development of MT in health settings and determining promising approaches for the use of MT technologies in health, found—based on 27 identified studies (2006 to 2016)—that the use of MT in health communication was an initial step to be followed by human editing [8]. It should be noted, however, that the included studies involved either statistical or rule-based MT systems, not neural machine translation, which is now predominant. The authors of the systematic review found that the included studies assessed various human and automatic MT evaluation methods. Translation performance was best for simple, less technical sentences and from English to Western European languages for most studied MTEs. The review pointed to continued accuracy concerns for the use of MT in health, "where excellent accuracy and a strong evidence base are critical." Another identified problem was the lack of shared evaluation criteria for MT [8].

The same author team developed a then-novel collaborative machine translation (MT) plus post-editing system, called PHAST (Public Health Automatic System for Translation,

phastsystem.org), for the production of multilingual educational materials in public health. They showed that it could become instrumental in the production of multilingual public health content while reducing the barriers of time and cost, and presented PHAST as "a new approach in public health informatics" that built its capacity on sharing translation resources via a groupware system to assure accuracy of MT through shared language expertise [9]. Cloud-based translation management systems, which offer workflow and translator management, web editors, translation memories, glossaries and machine translation, among other functionalities, and which allow flexible collaboration between machine translation and post-editing, have now become more common.

The challenges of translating clinical texts, particularly patient health records, have been researched, and various approaches have been tested, including neural machine translation [10]. Wu and colleagues assessed the statistical machine translation output of their in-house system and Google Translate in the biomedical domain from English into six other languages and the other way around. They used the common automated BLEU (Bi-Lingual Evaluation Understudy) metric and human judgment to evaluate MT quality, and reported high performance of their in-house system and Google Translate for three language—German, French and Spanish—translated into English and back [11]. Cochrane also participated in the three-year EU funded project "Health in my Language", which developed and evaluated health domain-adapted MT for consumer health information from English into Czech, German, Polish and Romanian. The project developed and tested a range of approaches to improve custom domain-adapted MT. The statistical and neural MT models that the project team developed and compared were not deemed reliable enough for Cochrane translations without post-editing, although user evaluation suggested that some people would prefer the domain-adapted MT over no translation at all. While there was some variation across languages, post-editing MT was typically quicker than translating without MT as a basis for Cochrane PLSs, and translators preferred post-editing MT [12].

Zilfiqar et al. tested Google Translate and DeepL for German to English translations, converting German scientific literature into English. The authors compared human and machine translations for complex sentences from old literature and a recent publication as a benchmark. They concluded that human care and intuition should be used before relying on machine translation of methods sections [6]. A researcher affiliated with the Cochrane France team (who is leading Cochrane's French translation activities) pre-translated a corpus of Cochrane review abstracts with DeepL and had them post-edited by translation master students to assess lexicogrammatical patterns for potential distortions. She showed that DeepL created specific sources of distortion of the translated patterns and argued for the need for special focus on these lexicogrammatical patterns in the post-editing efforts, in order to improve the quality of machine translation of medical texts [13].

Das and colleagues tested Google Translate to translate the safety guidelines of the American Academy of Pediatrics (AAP) into the 20 most commonly spoken languages in the United States, including Russian [14]. The authors evaluated the accuracy of the Google Translate results by performing human back-translation and assessing its quality using a five-point rubric adapted from the American Translators Association. They found that Google Translate did not meet the professional standard (5.00 points) for all but one language, Spanish, with Portuguese following (4.95 and 4.33, respectively). Russian Google translations scored 3.71, falling into the "acceptable" category. The authors argued that their findings demonstrated the inaccuracy of a popular machine translation service in translating the AAP safety guidelines and pointed out that inaccurate translation of medical texts may pose significant risks to consumer health.

We did not find any other research assessing machine translation for Russian translations of medical texts designed in plain language for the general public. To the best of our knowledge, our research is the first contribution that compares machine translation engines by measuring quantitative human post-editing efforts in an established translation workflow and quality assurance process using a randomized study design.

## 2. Materials and Methods

We developed our study, applying best practices of our main field, clinical research, where possible, including: prior literature search to assess the current state of research, development of a study protocol prior to study initiation, and randomized study design [15–18].

### 2.1. Preparation and Protocol Development

Prior to study initiation, we reviewed documentation of Memsource Machine Translation Quality Estimation (MTQE) and analysis features, developed the study protocol including the main steps and actions (Supplementary S1), discussed details and technical specifics within the author team, and prepared step-by-step instructions for volunteer participants in the Russian language.

### 2.2. Memsource Features

Memsource is a cloud-based translation management system, which allows users to set up complex translation workflows with numerous steps and assign translators and editors to different steps. It includes a segment-level bilingual editor, translation memories (TM) and glossaries. A segment typically constitutes a sentence or a phrase. Memsource can be connected to external systems—via an application programming interface (API) or one of its built-in connectors—in order to facilitate automatic data flow to their clients' systems or integration of tools, including MTEs.

Texts can be imported into Memsource and pre-translated, with a choice or combination of integrated MTEs and translation memories. For example, a translator could choose to pre-translate a text with existing translations from a TM as a priority where available, and with an MTE for any segments that do not have a match in the translation memory.

MTQE is an AI-powered feature that provides segment-level quality estimates for MT suggestions in the form of percentages similar to translation memory (TM) matches. These estimates give an initial indication of how much editing might be required for a given segment [19]. For example, if a text is translated with an MTE and the MTQE estimate for a given segment is 80%, a translator will likely edit 20% of that segment. If the MTQE estimate for a segment is 100%, a translator will likely not need to edit that segment at all.

Analyses can be generated on Memsource at different stages of the translation workflow to obtain data about the number of words, machine translation and translation memory matches, or the number of edits in translated texts. These can be broken down by complete text, segment, word, character or percentage of the text [20]. For example, one could run an analysis, after pre-translating a text, to obtain data about how many words or segments of the text were pre-translated with TM matches or MTE, or what percentage of the text had certain MTQE estimates. One could also run an analysis, after editing a text, to obtain data on how many characters, words, or segments were edited—or what percentage of the text was edited.

Screenshots illustrating Memsource pre-translation, analyses and MTQE in the Memsource editor are available in Supplementary S2.

### 2.3. MTE for Comparison

We compared three commercial MTEs: DeepL, Google Translate and Microsoft Translator, all of which were integrated into Memsource via application programming interface (API) and were available at relatively low-cost subscription fees (based on the number of translated characters). All three MTEs are domain-generic and use a neural machine translation approach. Their algorithms and source codes are closed-source and not publicly available, and they use a combination of methodologies and sources of data to develop their systems on an ongoing basis. Therefore, they may deliver different output and quality at different points in time without users necessarily noticing when changes occur.

DeepL is a relatively young neural machine translation service launched in 2017 and trained on the Linguee bilingual corpora database created by the same company. It

supports translation between 11 (mainly European) languages as of January 2021. The type of neural network and architecture DeepL uses has been said to lead to more natural sounding translations than those of its competitors [21].

Google Translate was launched in 2006 as a statistical machine translation service, but has since transitioned to neural machine translation. It supports 109 languages as of January 2021 [22].

Microsoft Translator first launched as a public tool in 2007, and has been available as a neural machine translation service since 2018. It supports more than 70 languages [23].

Cochrane has very limited resources for translation and their core work is health research, not translation or translation research. It was therefore not reasonable to test an open-source MTE within this study.

### 2.4. Principle Approach of the Study and Evaluation Method

Our study aimed to replicate—as much as possible—our standard translation environment, workflow, and content. We compared the three MTEs by pre-translating the same number of Cochrane plain language summaries (PLSs) with each MTE within Memsource, and then having them post-edited twice by select volunteer translators and editors following standard procedures. This meant that post-editors were familiar with the system, processes, and type of content. We used Memsource MTQE and analysis features as described above to record initial AI-estimates of required post-editing effort following pre-translation, as well as precise recording and numerical presentation of the amount of human editing required for each MTE after post-editing.

In principle, our method was similar to the translation edit (or error) rate (TER) metric, which measures the amount of editing that a human would have to perform to edit MT output, so that it exactly matches a reference gold-standard human translation [24]. Within our approach, the final translation constituted the gold standard, and instead of having to compute the editing distance, we obtained the data directly from Memsource.

We analyzed and interpreted the obtained data to compare the performance of the three MTEs as outlined below.

### 2.5. Description of the Dataset, Selection and Randomization of Plain Language Summaries (PLSs)

We obtained a list of 199 Cochrane plain language summaries (PLSs), which had been published on the Cochrane Library over the previous 12 months (from May 2018 to April 2019), and which had not been published in Russian at that time. We excluded any PLSs that were already in the regular translation workflow ($n$ = 28), and then randomly selected 90 PLSs using the online randomization tool available from https://www.random.org/lists/ (accessed on 30 December 2020).

We used 90 PLSs, not 100 as planned in the study protocol, so that we could distribute them evenly across the 3 MTEs. We randomized the 90 PLSs into three sets of 30 PLSs for each of the three MTEs. We pre-translated the PLSs within Memsource with the assigned MTE, as per randomization, to obtain 30 PLSs translated by DeepL, 30 by Google Translate and 30 by Microsoft Translator, respectively.

Each Cochrane review and its associated PLS is produced by a different international author team. Reviews cover many health topics and vary in the complexity of the interventions and conditions they address. Cochrane has rigorous methodological standards and processes for its review production, including for its PLSs; however, given their heterogenous subject matter and authorship, the PLSs themselves are also found to be heterogenous in length, language, and other aspects of quality [25]. This is the reality that Cochrane translators deal with, and our dataset aimed to reflect that reality. We assessed the heterogeneity of our dataset by calculating average word count per PLS assigned to each MTE as arithmetic means with standard deviations and medians with interquartile range as a measure of dispersion, to show that the randomized design resulted in a statistically homogenous set of 30 PLSs per MTE in relation to word count. Average word counts are presented in Table 1. Overall, the data did not show statistically significant differences in

word count (asymptotic 2-sided significance = 0.178). It was therefore unlikely that the PLS length impacted our findings.

**Table 1.** Average word count per plain language summary (PLS) per machine translation engine (MTE) as arithmetic means with standard deviations and medians with interquartile range.

| Measure (Rounded Values) | DeepL | Google Translate | Microsoft Translator |
|---|---|---|---|
| Median (Q1/Q3) | 433 (365/546) | 455 (389/574) | 523 (402/594) |
| Mean (SD) | 450 ($\pm$134) | 484 ($\pm$123) | 510 ($\pm$123) |

*2.6. Participants*

We invited 10 experienced Russian volunteer translators and editors, all of whom had demonstrated efficiency and high quality in translation and/or editing over the last 5 years, to post-edit the machine translated PLSs. Two translation project managers reviewed and edited the post-edited machine translations in a second editing step.

*2.7. Analyses and Steps of the Study*

To obtain the data estimating or measuring post-editing effort of the MTEs, we created three analyses in Memsource:

- Default analysis, to obtain MTQE-automated estimates of post-editing effort following pre-translation;
- Post-editing 1, to obtain calculations of the amount of post-edited text following the first editing step; and
- Post-editing 2, to obtain calculations of the amount of post-edited text in total following the second editing step (including any edits made in the first editing step).

We realized that the study could not be completed within 1 month, as specified in the protocol, due to its complexity. We therefore allowed more time for each planned step and completed the study over a 2-month-period. We omitted the optional step 7 described in the protocol.

2.7.1. Default Analysis

We conducted pre-translation (from English to Russian) in Memsource, then pre-populated all 90 PLSs with TM matches as a priority where available and machine translation as a secondary priority, mimicking the standard Russian translation workflow.
Furthermore, at this step:

- We generated the Default analysis in Memsource for all 90 PLSs to obtain the MTQE figures estimating the editing effort of the initial MT output.
- We randomized the PLSs into smaller sets for 10 post-editors: 3 PLSs per MTE for each post-editor, or 9 PLSs per post-editor in total.
- We assigned PLSs to the 10 post-editors according to randomization, and we aimed to "blind" the post-editors: we did not inform participants which MTE was used to pre-translate each PLS to avoid potential bias. However, they could have found this information in Memsource if they had looked for it; we were not able to reliably conceal that information from the post-editors.

2.7.2. Post-Editing 1 Analysis

10 volunteers post-edited the pre-translated PLSs that had been assigned to them. We then generated the Post-editing 1 analysis, which allowed us to perform numerical analysis of the amount of editing undertaken by post-editors for each PLS and MTE.

2.7.3. Post-Editing 2 Analysis

At this step:

- We randomized all 90 PLSs anew, into two sets of 45 PLSs, for two final editors.
- 2 translation project managers performed a second and final round of post-editing of all 90 PLSs.

We then generated the Post-editing 2 analysis, consisting of the total amount of all editing work undertaken for each PLS and MTE, including the initial and second post-editing.

Obtained analyses for each PLS at each step are available in Supplementary S3–S5. The steps of the study are illustrated in a flow diagram (Figure 1).

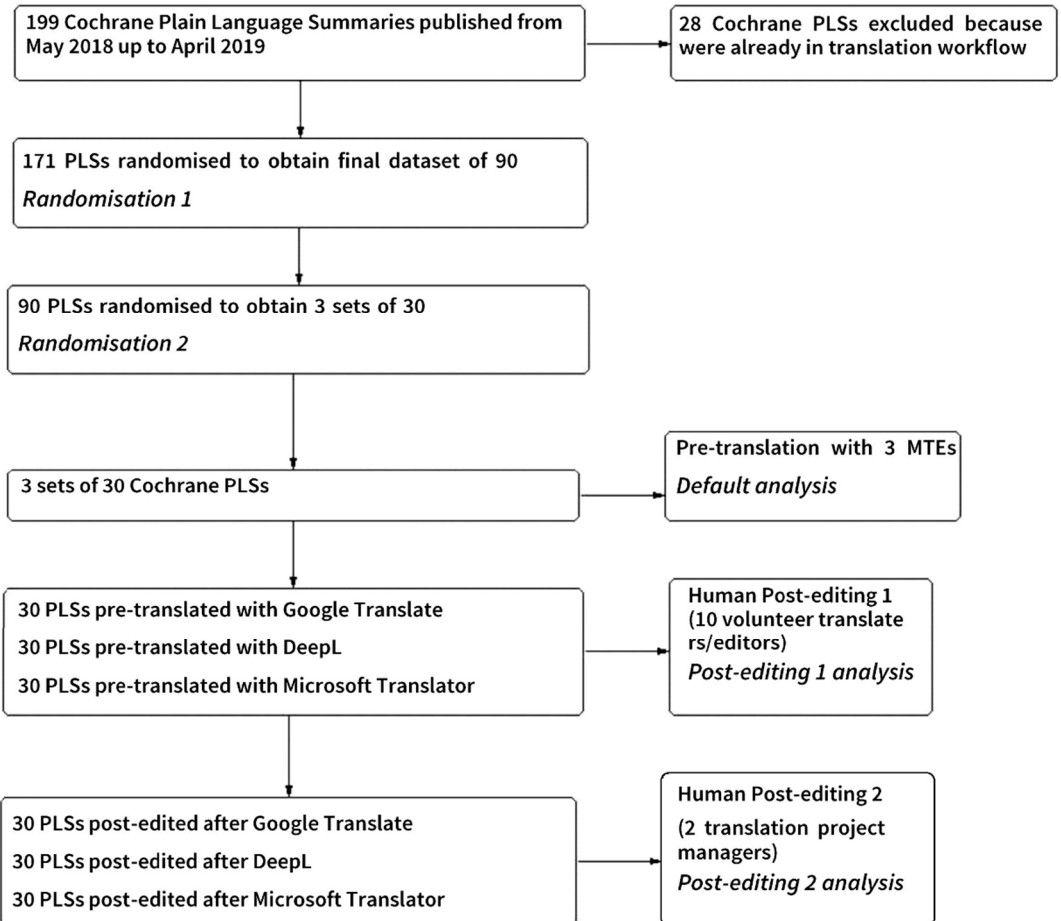

**Figure 1.** Flow diagram illustrating the different steps of the study. Notes: PLSs—plain language summaries; MTEs—machine translation engines.

We combined the data of each of the analyses from all PLSs by MTE to obtain three sets of results (made up of 30 analyses per MTE). We combined the percentage of MT and TM matches in each quality estimate range (for the Default analysis), in each range of edits (for post-editing analyses) and for each MTE separately. We then calculated the average percentages as arithmetic means and medians. The medians were compared using the Kruskal–Wallis H test with SPSS (SPSS Inc., Chicago, IL, USA), including a pairwise post hoc test for statistical significance. The predefined significance threshold was $p < 0.05$.

We provide examples of original English and post-edited Russian PLSs in Supplementary S6.

## 3. Results

### 3.1. Default Analysis

For the Default analysis, we analyzed the estimated quality percentages generated by MTQE by MTE. The Default analysis distributed percentages into the following ranges: 100%, 95–99%, 85–94%, 75–84%, 50–74% and 0–49%.

A higher percentage range denoted a better estimated MT quality. More specifically, the different quality estimate ranges of the Default analysis constituted the following:

- The 100% percentage range represents the highest possible quality estimate, and suggests that text in this range may require no editing. The MTE with the highest percentage of text in this range is perhaps the best.
- The 95–99% percentage range indicates very high-quality estimates, and suggests that text in this range may require only minor editing—perhaps only 1–5% of the text. The MTE with the highest percentage of text in this range is perhaps the best.
- The 85–94% percentage range presents texts with fairly high-quality estimates that require some editing; 6–15% of the text may need editing.
- The 75–84% percentage range indicates good- or moderate-quality estimates that require between 16–25% of text to be edited—or up to a quarter of the text.
- The 50–74% percentage range includes text with poor-quality estimates, indicating that 26–50% of the text may need editing—or up to half of the text. The MTE with the highest percentage of text in this range is perhaps the worst.
- The 0–49% percentage range represents low-quality estimates that require substantial editing; 51–100% of the text may require editing—up to the entire text. The MTE with the highest percentage of text in this range is perhaps the worst.

The Default analysis for each PLS showed what percentage of text had MT quality scores ranging from 0% to 100%. The analysis showed the percentage of machine-translated text, as well as TM matches for each of the predefined quality estimate ranges (100%, 95–99%, 85–94%, 75–84%, 50–74%, 0–49%).

We combined the data of the Default analysis from all PLSs by MTE to obtain three sets of Default analyses (made up of 30 analyses per MTE). We combined the percentage of MT and TM matches in each quality estimate range and for each MTE separately, and then calculated the average percentages as arithmetic means and medians (Table 2).

Google Translate had the highest average percentages of text in the quality-estimate ranges of 100% and 75–84%. There were no differences between the three MTEs for average percentages of text in the quality estimate ranges of 95–99% and 85–94%, due to low or near-zero values. In the quality estimate range of 50–74%, DeepL performed worse than both Google Translate and Microsoft Translator, whereas in the quality estimate range of 0–49%, Microsoft Translator performed worse than both DeepL and Google Translate.

The results of the Default analyses suggested that Google Translate showed the best results. DeepL performed a little worse than Google Translate, and Microsoft Translator showed the worst results overall.

### 3.2. Post-Editing 1 Analysis

The Post-editing 1 analysis was generated after the first series of human editing of pre-translated content by volunteer translators and editors.

The post-editing analysis represented the human editing effort; in other words, how much of the machine-translated text was edited. The analysis classified edited text into percentage ranges equivalent to the Default analysis: 100%, 95–99%, 85–94%, 75–84%, 50–74% and 0–49%.

**Table 2.** Results of Default analysis per MTE—the average percentages of machine translation (MT) and translation memories (TM) per estimated quality range. TM matches are included in the table for completeness, but they were not relevant for our assessment.

| MTQE Range | | DeepL | Google Translate | Microsoft Translator |
|---|---|---|---|---|
| | | % of MT or TM text in each range [1] | % of MT or TM text in each range [1] | % of MT or TM text in each range [1] |
| All | | 100 (100 [100–100]) | 100 (100 [100–100]) | 100 (100 [100–100]) |
| Repetitions | | 1.18 (0 [0–2.3]) | 0.75 (0 [0–1.15]) | 0.33 (0 [0–0]) |
| 101% TM | | 0.67 (0 [0–0]) | 0 (0 [0–0]) | 0 (0 [0–0]) |
| 100% (no edits expected) | TM | 4.66 (3.10 [2.2–6.1]) | 4.17 (2.65 [1.9–3.3]) | 3.47 (3.25 [2.1–4.1]) |
| | NT | 0 (0 [0–0]) | 0 (0 [0–0]) | 0 (0 [0–0]) |
| | MT | 0.03 (0 [0–0]) | 0.09 (0 [0–0]) | 0.01 (0 [0–0]) |
| 95–99% (1–5% of text to be edited) | TM | 1.19 (0 [0–0.5]) | 0.73 (0 [0–1.3]) | 1.10 (0 [0–1.1]) |
| | NT | 0.03 (0 [0–0]) | 0.03 (0 [0–0]) | 0 (0 [0–0]) |
| | MT | 0 (0 [0–0]) | 0 (0 [0–0]) | 0 (0 [0–0]) |
| 85–94% (6–15% of text to be edited) | TM | 0 (0 [0–2]) | 0 (0 [0–1.5]) | 0 (0 [0–0]) |
| | MT | 0 (0 [0–0]) | 0 (0 [0–0]) | 0 (0 [0–0]) |
| 75–84% (16–25% of text to be edited) | TM | 1.94 (0 [0–3]) | 2.18 (0.35 [0.0–3.9]) | 1.31 (0 [0–1.7]) |
| | MT | 35.70 (35.95 [26.1–44.1]) | 37.92 (39.5 [24.2–54.2]) | 33.37 (31.65 [21.7–43.5]) |
| 50–74% (26–50% of text to be edited) | MT | 6.04 (3.55 [1.1–7.3]) | 4.00 (3.4 [0.8–6.8]) | 4.78 (2.7 [0.8–7.9]) |
| 0–49% (51–100% of text to be edited) | MT | 47.00 (48.15 [37.3–59.3]) | 48.87 (48.15 [36.2–59.5]) | 54.82 (55.15 [44.8–64.2]) |

Notes: [1] in brackets: median [1st Quartile (Q1)–3rd Quartile (Q3)]; TM—translation memory; NT—non-translatables; MT—machine translation; Colors: green—best performing MT in this range; yellow—second best performing MT; red—worst performing MT.

The more text the post-editors modified, the lower the indicated percentage range—and supposedly the poorer the MT output. More specifically, the different percentage ranges of edits in the Post-editing 1 analysis were organized as follows:

- The 100% percentage range indicates that no edits were made to the text. The MTE with the highest percentage of text in this range is presumably the best.
- The 95–99% percentage range means that hardly any edits were made by human post-editors; only 1–5% of the text was edited. The MTE with the highest percentage of text in this range is perhaps the best.
- The 85–94% percentage range indicates that between 6–15% of the text was edited.
- The 75%–84% percentage range means that between 16–25% of the text was edited—or up to a quarter of the text.
- The 50–74% percentage range indicates that between 26–50% of the text was edited—or up to a quarter of the text. The MTE with the highest percentage of text in this range is perhaps the worst.
- The 0–49% percentage range means that between 51–100% of the text was edited—or up to the entire text. The MTE with the highest percentage of text in this range is probably the worst.

We combined the data of the Post-editing 1 analysis from all PLSs by MTE to obtain three sets of Post-editing 1 analyses (made up of 30 analyses per MTE). We combined the percentage of MT and TM matches in each range of edits (and for each MTE separately) and calculated the average percentages as arithmetic means and medians (Table 3).

**Table 3.** Results of Post-editing 1 analysis per MTE—the average percentage (median) of post-edited MT and TM matches per range of edits.

| Percentage Range of Edits | DeepL | | | Google Translate | | | Microsoft Translator | | |
|---|---|---|---|---|---|---|---|---|---|
| | % of MT or TM text in each range of edits [1] | | | % of MT or TM text in each range of edits [1] | | | % of MT or TM text in each range of edits [1] | | |
| | TM | MT | All | TM | MT | All | TM | MT | All |
| All | 9.22 (6.35 [4.4–11.1]) | 90.78 (93.65 [88.9–95.6]) | 100 (100 [100–100]) | 6.46 (4.9 [3.9–7.6]) | 93.54 (95.1 [92.4–96.1]) | 100 (100 [100–100]) | 6.16 (5.6 [3.9–7.6]) | 93.84 (94.35 [92.4–96.1]) | 100 (100 [100–100]) |
| Repetitions | 1.18 (0 [0.0–2.3]) | - | 1.18 (0 [0.0–2.3]) | 0.75 (0 [0.0–1.5]) | - | 0.75 (0 [0.0–1.5]) | 0.31 (0 [0–0]) | - | 0.31 (0 [0–0]) |
| 100% (no edits) | 4.49 (2.9 [2.3–5.6]) | 12.95 (11.65 [5.1–18.1]) # | 17.44 (14.35 [10.5–26.5]) # | 3.25 (2.3 [1.7–3.0]) | 17.82 (15.2 [4.1–28.7]) # | 21.08 (18.35 [6.0–31.2]) # | 3.2 (3.25 [2.1–4.2]) | 4.73 (3.4 [0.6–6.6]) # | 7.93 (6.9 [4.7–10.4] # |
| 95–99% (1–5% of text edited) | 0.27 (0 [0–0]) | 2.64 (0 [0–0]) | 2.91 (0 [0.0–3.4]) | 0.12 (0 [0–0]) | 2.96 (0 [0.0-6.6]) | 3.08 (0 [0.0–6.6]) | 0 (0 [0–0]) | 1.12 (0 [0.0–2.2]) | 1.12 (0 [0.0–2.2]) |
| 85–94% (6–15% of text edited) | 0.52 (0 [0–0]) | 16.82 (14.5 [8.9–25.9]) # | 17.33 (15.05 [8.9–25.9]) # | 0.44 (0 [0–0]) | 20.98 (20.05 [7.2–30.2]) # | 21.43 (20.45 [10.5–30.2]) # | 0.02 (0 [0–0]) | 6.19 (6.5 [0.0–10.5]) # | 6.20 (6.5 [0.0–10.5]) # |
| 75–84% (16–25% of text edited) | 0.76 (0 [0.0–0.7]) | 16.08 (16.3 [7.0–20.3]) | 16.83 (16.55 [7.0–23.5]) | 0.43(0 [0.0–0.4]) | 15.99 (16.9 [9.6–23.1]) | 16.43 (16.9 [9.7–24.2]) | 0.34 (0.00 [0.0–0.6]) | 13.49 (12.25 [6.2–20.4]) | 13.83 (13.2 [7.2–20.4]) |
| 50–74% (26–50% of text edited) | 1.7 (0.7 [0.0–1.8]) | 33.37 (31.95 [17.5–49.6]) | 35.06 (37.15 [17.5–50.4]) | 1.01 (0.55 [0.0–0.8]) | 27.79 (27.2 [10.0–44.2]) # | 28.79 (28.65 [10.7–44.2]) # | 1.21 (0.50 [0.0–2.2]) | 41.35 (38.65 [32.9–52.6]) # | 42.56 (39.3 [34.5–55.0]) # |
| 0–49% (51–100% of text edited) | 0.31 (0 [0.0–0.6]) | 8.92 (7.2 [1.3–12.7]) # | 9.23 (7.55 [2.0–12.7]) # | 0.44 (0 [0.0–0.7]) | 8.01 (4.7 [0.0–10.9]) # | 8.44 (5.25 [0.0–10.9]) # | 1.06 (0.25 [0.0–1.8]) | 26.97 (25.55 [9.8–41.6]) # | 28 (25.85 [9.8–43.4]) # |

Notes: [1] in brackets: median [1st Quartile (Q1)–3rd Quartile (Q3)]; TM—translation memory; MT—machine translation; NT—non-translatables: both mean and median equal zero across all comparisons and ranges of edits, not shown in the table to allow space; #—$p < 0.05$ between any of the three MTE in each range of edits (across one horizontal line of the table), # is placed in those cells of the table for which significant differences were detected in pairwise comparisons. Colours: green—best performing MT in this range; yellow—second best performing MT; red—worst performing MT.

Google Translate showed the best results in most ranges of edits in comparison with DeepL and Microsoft Translator, so Google Translate arguably performed best overall. DeepL performed a little worse than Google Translate. Microsoft Translator had the worst results. The results of the Post-editing 1 analysis corresponded to the results of the Default analysis.

### 3.3. Post-Editing 2 Analysis

The Post-editing 2 analysis was generated after the second series of human editing. It followed the same patterns as the Post-editing 1 analysis, and the interpretation of different percentage ranges of edits in the Post-editing 2 analysis is equivalent to that of the Post-editing 1 analysis: the higher the percentage range, the more edits were made to the text. The averaged results in Table 4 show how much the machine-translated text by each MTE was edited in the two subsequent editing steps.

Google Translate had the highest average percentage of text in the 100% and 85–94% ranges; it also had the lowest average percentage of text the 50–74% and 0–49% ranges. According to our interpretation of the ranges of edits, this means that Google Translate is perhaps the best among the three compared MTEs.

DeepL showed reasonable results in the percentage ranges of 100%, 85–94%, and 0–49%, and had the highest average percentage of text in the 95–99% range, though its advantage was not statistically significant.

Microsoft Translator had the lowest average percentage of text in the 100%, 95–99% and 85–94% ranges, and the highest average percentage of text in the 50–74% and 0–49% ranges, which means that Microsoft Translator is probably the worst, in comparison with the other two MTEs.

**Table 4.** Results of Post-editing 2 analysis per MTE—the average percentage (median) of post-edited MT and TM matches per range of edits after two editing steps.

| Percentage Range of Edits | DeepL | | | Google Translate | | | Microsoft Translator | | |
|---|---|---|---|---|---|---|---|---|---|
| | % of MT or TM text in each range of edits [1] | | | % of MT or TM text in each range of edits [1] | | | % of MT or TM text in each range of edits [1] | | |
| | TM | MT | All | TM | MT | All | TM | MT | All |
| All | 10.23 (8.8 [5.0–12.2]) | 89.77 (91.2 [87.8–95.0]) | 100 (100 [100–100]) | 6.99 (5.25 [4.1–9.0]) | 93.01 (94.75 [91.0–95.9]) | 100 (100 [100–100]) | 6.43 (5.80 [3.9–7.9]) | 93.57 (94.20 [92.1–96.1]) | 100 (100 [100–100]) |
| Repetitions | 1.18 (0 [0.0–2.3]) | 0 | 1.18 (0 [0.0–2.3]) | 0.75 (0 [0.0–1.5]) | 0 | 0.75 (0 [0.0–1.5]) | 0.33 (0 [0–0]) | 0 | 0.33 (0 [0–0]) |
| 100% (no edits) | 3.90 (2.85 [2.1–5.1]) | 6.82 (6.2 [1.9–9.3]) # | 10.72 (9.65 [5.1–12.2]) # | 3.28 (2.20 [1.4–3.1]) | 11.2 (6.25 [2.5–22.7]) # | 14.51 (12.25 [4.2–24.3]) # | 3.02 (2.80 [2.1–3.8]) | 2.05 (1.5 [0.0–2.9]) # | 5.07 (4.5 [3.3–6.4]) # |
| 95–99% (1–5% of text edited) | 0 (0 [0–0]) | 1.31 (0 [0–0]) | 1.31 (0 [0–0]) | 0.12 (0 [0–0]) | 1.29 (0.00 [0.0–2.2]) | 1.41 (0.00 [0.0–2.8]) | 0 (0 [0–0]) | 0.70 (0 [0–0]) | 0.70 (0 [0–0]) |
| 85–94% (6–15% of text edited) | 0.42 (0 [0–0]) | 11.83 (10.9 [6.9–18.2]) # | 12.25 (10.9 [6.9–18.7]) # | 0.39 (0.00 [0.0–0.0]) | 13.63 (12 [7.1–19.4]) # | 14.02 (12.75 [7.1–19.4]) # | 0 (0 [0–0]) | 4.23 (3.65 [0.0–5.9]) # | 4.23 (3.65 [0.0–5.9]) # |
| 75–84% (16–25% of text edited) | 0.95 (0.00 [0.0–1.3]) | 16.84 (17.3 [9.2–24.3]) # | 17.79 (17.8 [9.2–24.3]) # | 0.53 (0 [0.0–0.7]) | 19.65 (19.05 [9.8–29.8]) # | 20.18 (19.05 [9.8–29.8]) # | 0.53 (0 [0.0–1.0]) | 10.41 (9.55 [4.5–14.6]) # | 10.94 (9.6 [5.0–15.5]) # |
| 50–74% (26–50% of text edited) | 3.16 (1.3 [0.7–2.1]) | 41.23 (43.55 [30.7–54.9]) | 44.39 (45.85 [33.2–57.9]) | 1.42 (0.8 [0.0–1.5]) | 35.7 (35.45 [25.1–46.9]) | 37.12 (40.15 [25.6–48.7]) | 1.52 (0.80 [0.0–2.8]) | 42.17 (43.6 [35.0–50.4]) | 43.69 (44.30 [35.9–52.8]) |
| 0–49% (51–100% of text edited) | 0.62 (0 [0.0–0.8]) | 11.74 (10.25 [4.4–16.2]) # | 12.37 (10.25 [5.3–17.0]) # | 0.48 (0 [0.0–0.4]) | 11.52 (7.85 [4.7–14.2]) # | 12.00 (8.85 [5.0–15.0]) # | 1.03 (0 [0.0–1.5]) | 34.00 (34.25 [21.9–44.2] # | 35.03 (34.25 [22.2–45.3]) # |

Notes: [1] in brackets: median [1st Quartile (Q1)–3rd Quartile (Q3)]; TM—Translation memory; MT—machine translation; NT—non-translatables: both mean and median equal zero across all comparisons and ranges of edits, not shown in the table to allow space; #—$p < 0.05$ between any of the three MTE in each range of edits (across one horizontal line of the table), # is placed in those cells of the table for which significant differences were detected in pairwise comparisons. Colours: green—best performing MT in this range; yellow—second best performing MT; red—worst performing MT.

## 4. Discussion

We developed a systematic and randomized approach to comparing the performance of different MTEs with a focus on assessing human post-editing effort. Figure 2 illustrates the total amount of edits per MTE following the two consecutive editing steps (post-editing analysis 2) and the advantage of Google Translate and DeepL over Microsoft Translator.

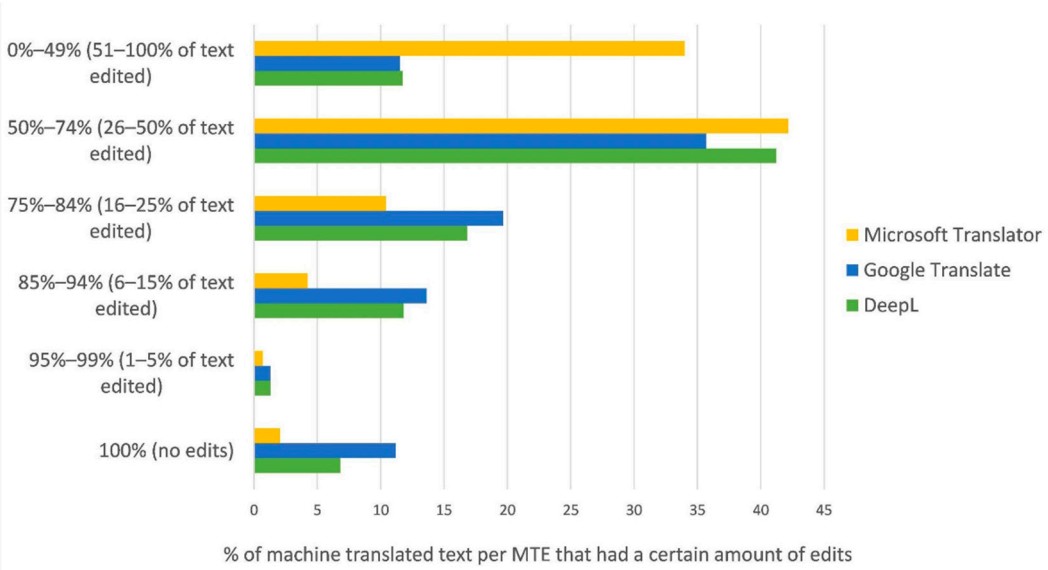

**Figure 2.** Percentage of machine translated text per MTE after two consecutive human post-editing steps. Google Translate and DeepL are shown to have had more MT output with fewer edits (no edits or up to 25% of text edited), whereas Microsoft Translator had more MT output that required more substantial edits.

Our approach did not assess the types of mistakes the MTEs generated or the types of edits that were made by the human editors; for example, to what extent they were correcting grammatical errors, terminology, or style. Instead, it measured the overall editing effort required using different MTEs to achieve an established quality standard in a set quality assurance process and environment—the gold standard established by Cochrane Russia for Russian translations of Cochrane PLSs. While the approach was fairly intensive in terms of human resources, as compared to automated MT evaluation methods such as BLEU, it allowed us to specifically assess which MTE would best contribute to efficient translation in our established setup. The same approach would not work to assess distinct quality aspects of MTEs such as grammar, terminology or style—and would have limited application in a context without established quality assurance processes. However, our approach could be applied by other translation teams with established quality assurance processes in any domain to select the most appropriate MTE for their context.

It should be noted that our approach relied on us having access to certain features in Memsource, and Memsource requires a paid subscription. People wanting to replicate our approach would need to have access to tools that allow them to calculate post-editing effort in a similar way.

Given that the results from the analysis of AI-powered MTQE were in line with the results of our human post-editing analyses, it may be sufficient, or at least a good start, to use such automated tools to determine the most appropriate MTE for specific content.

Our approach and findings contribute to further development of the understanding of the role of machine translation of medical content as the initial step preceding human editing, with full responsibility resting on the human post-editing step for the overall accuracy and quality of the resultant final translation—as evidenced by the systematic review of Dew and co-authors [8].

Our findings are also echoed by the results of Zilfiqar et al., who tested Google Translate and DeepL for German to English translations of scientific texts and found them to be reliable, yet in need of human care and intuition to guarantee accurate translation [6].

Overall, our findings on MTE performance seem largely in line with other research on the current role of machine translation for medical content as an initial step preceding human editing in most languages, including Russian, but not as a standalone, reliable translation approach that can achieve the required accuracy and quality [8,14].

Future advances in MT technologies, language development, language synergy processes and the MT industry may change the scope of research in this field. We plan to repeat our analyses in a couple of years, as different MT systems and evaluation methods become available and affordable for low-resource settings.

## 5. Conclusions

Among the three MTEs that we tested, Google Translate required the least editing and appeared to perform best for Russian translations of Cochrane PLSs, while DeepL also showed good results. Microsoft Translator performed worse than DeepL and Google Translate. At this point in time, we would recommend Google Translate, with DeepL as the second-best option, for machine translation of Cochrane PLSs into Russian.

While Google Translate performed slightly better than DeepL, we have opted to use DeepL as default MT engine in our translation workflow, as DeepL offers preferable IP and copyright terms. DeepL is based in Germany and complies with European GDPR regulations. Subscribed customers of its service retain the rights to their own content and grant a nonexclusive license to DeepL, as required for it to provide its services to the customer [26], while Google's terms of service apply to users of Google Translate, and vary by country.

**Supplementary Materials:** The protocol of the study and Individual analyses data are available online at https://www.mdpi.com/2227-9709/8/1/9/s1, **S1**: Protocol of a study: Assessing human post-editing effort to compare performance of three machine translation engines for English to Russian translation of Cochrane plain language health information. **S2:** Screenshots of Memsource pre-translation feature, editor with Machine Translation Quality Estimates and Translation Memory matches, and post-editing analysis feature. Included with permission from Memsource. **S3**: Individual analysis data for each of the 30 Cochrane plain language summaries, pre-translated with DeepL, downloaded from Memsource and archived as World tables. **S4:** Individual analysis data for each of the 30 Cochrane plain language summaries, pre-translated with Google Translate, downloaded from Memsource and archived as World tables. **S5**: Individual analysis data for each of the 30 Cochrane plain language summaries, pre-translated with Microsoft Translator, downloaded from Memsource and archived as World tables. **S6:** Samples of original English and translated Russian plain language summaries.

**Author Contributions:** Conceptualization, L.E.Z. and J.R.; methodology, L.E.Z., E.V.Y. and J.R.; software, J.R.; validation, L.E.Z., E.V.Y. and J.R.; formal analysis, E.V.Y., A.I.G. and L.E.Z.; investigation, E.V.Y., A.I.G. and L.E.Z.; resources, L.E.Z., E.V.Y. and J.R.; data curation, A.I.G.; writing—original draft preparation, L.E.Z. and E.V.Y.; writing—review and editing, L.E.Z. and J.R.; visualization, E.V.Y. and A.I.G.; supervision, L.E.Z. and J.R.; project administration, L.E.Z. and E.V.Y.; funding acquisition, L.E.Z. All authors have read and agreed to the published version of the manuscript.

**Funding:** This research received no external funding.

**Institutional Review Board Statement:** Not applicable.

**Informed Consent Statement:** Not applicable.

**Data Availability Statement:** Data is contained within the article. Individual analysis data for each of the included 90 Cochrane plain language summaries was obtained from Memsource, its Analysis feature, available to authors through Cochrane paid subscription. Downloaded individual analyses data in tabulated format in Word files have been archived in Supplementary Materials S3–S5.

**Acknowledgments:** We would like to thank the volunteer post-editors of our study—Mikhail E. Kukushkin, Dilyara F. Nurkhametova, Ayrat U. Ziganshin, Dina A. Lienhard, Aelita A. Kamalova, Alina V. Ivanyuk, Olga A. Goluchenko, Karyna G. Uvarova—for their participation, which made our study possible. We would like to acknowledge Judith Deppe, Cochrane Multi-language Programme Manager, for administrative and logistical support.

**Conflicts of Interest:** The authors declare no conflict of interest.

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
