# Peer review of "Assessing Human Post-Editing Efforts to Compare the Performance of Three Machine Translation Engines for English to Russian Translation of Cochrane Plain Language Health Information: Results of a Randomised Comparison"

_informatics, doi:10.3390/informatics8010009_

Round 1
Reviewer 1 Report
I find this text extremely clear and well written. I found the text enjoyable to read and I found its findings informative. The results of studies of this nature, though fairly ephemeral, are useful, not only to those who develop MT systems, but also those who make use of them in research and professional contexts. The study is not hugely ambitious in scope. However, it is well executed and is easily transferable to different language pairs or knowledge domains, giving substantial opportunity for follow-up research.
My only criticism is that the prose occasionally slips into somewhat casual phraseology, which does not serve the otherwise strong writing very well. Therefore, I would recommend that the authors reread the text for style, and replace the small number of contractions in the text with full words (e.g. "wouldn't" > "would not", "didn't" > "did not", "wasn't" > "was not").
Author Response
Cover / rebuttal letter to Reviewer 1
Dear editors and reviewers,
With this letter we thank you for the detailed peer-review process and helpful suggestions for the improvement of our manuscript “Assessing human post-editing effort to compare performance of three machine translation engines for English to Russian translation of Cochrane plain language health information: results of a randomised comparison” by Liliya Eugenevna Ziganshina, Ekaterina V. Yudina, Azat I. Gabdrakhmanov, and Juliane Ried.
We are uploading the revised version to the Editorial manager.
Our answers to reviewers’ comments and suggestions:
Reviewer 1
“I find this text extremely clear and well written. I found the text enjoyable to read and I found its findings informative. The results of studies of this nature, though fairly ephemeral, are useful, not only to those who develop MT systems, but also those who make use of them in research and professional contexts. The study is not hugely ambitious in scope. However, it is well executed and is easily transferable to different language pairs or knowledge domains, giving substantial opportunity for follow-up research.
My only criticism is that the prose occasionally slips into somewhat casual phraseology, which does not serve the otherwise strong writing very well. Therefore, I would recommend that the authors reread the text for style, and replace the small number of contractions in the text with full words (e.g. "wouldn't" > "would not", "didn't" > "did not", "wasn't" > "was not").”
Authors’ answer:
Dear Reviewer 1,
Thank you very much for your positive assessment and appreciation of our manuscript and very helpful suggestions.
We have implemented the suggested changes fully and highlighted the changed word pairs in yellow.

Reviewer 2 Report
- The manuscript conducted a benchmarking analysis on three well-known machine translation engines (MTE) (DeepL, Google Translate, Microsoft Translator) for Russian translations of Cochrane Plain Language Summaries (PLSs) by assessing the quantitative human post-editing effort.
- There are many research papers in the literature conducting the benchmarking analysis on the reliability and quality of the auto machine translation tools. This manuscript applying it in the healthcare information process is interesting and unique, and could gain a lot of attentions to MT users.
- The manuscript is well written and organized. However, Section 0 (line 46-52) must be removed since the section is not a part of the transcript.
- The manuscript used three MTEs in its benchmarking comparison and analysis. Each MTE has its own unique vocabulary database/data structure and AI algorithms used to handle the translation. It is suggested that authors should specify the differences among them in the background/related work section.
- It is suggested that the dataset used for this research need to be explained and an example of comparison between the pre-translation of an original PLS (from English to Russian) and post-editing from a human translator should be provided to clearly illustrate the proposed approach.
Author Response
Cover / rebuttal letter to Reviewer 2
Dear editors and reviewers,
With this letter we thank you for the detailed peer-review process and helpful suggestions for the improvement of our manuscript “Assessing human post-editing effort to compare performance of three machine translation engines for English to Russian translation of Cochrane plain language health information: results of a randomised comparison” by Liliya Eugenevna Ziganshina, Ekaterina V. Yudina, Azat I. Gabdrakhmanov, and Juliane Ried.
We are uploading the revised version to the Editorial manager.
Our answers to reviewers’ comments and suggestions:
Reviewer 2
Points 1 – 3:
- “The manuscript conducted a benchmarking analysis on three well-known machine translation engines (MTE) (DeepL, Google Translate, Microsoft Translator) for Russian translations of Cochrane Plain Language Summaries (PLSs) by assessing the quantitative human post-editing effort.
- There are many research papers in the literature conducting the benchmarking analysis on the reliability and quality of the auto machine translation tools. This manuscript applying it in the healthcare information process is interesting and unique, and could gain a lot of attentions to MT users.
- The manuscript is well written and organized However, Section 0 (line 46-52) must be removed since the section is not a part of the transcript.”
Authors’ answer:
Dear Reviewer,
Thank you very much for your positive assessment and appreciation of our manuscript and very helpful suggestions.
We removed the lines 46-52 and apologise for leaving them in, these lines belong to the template.
Point 4:
“The manuscript used three MTEs in its benchmarking comparison and analysis. Each MTE has its own unique vocabulary database/data structure and AI algorithms used to handle the translation. It is suggested that authors should specify the differences among them in the background/related work section.”
Authors’ answer:
Dear Reviewer,
Thank you for this suggestion. We added short descriptions (characteristics) of the three MTEs into the Methods section, subsection MTE for comparison, in yellow highlight.
Point 5:
- It is suggested that the dataset used for this research need to be explained and an example of comparison between the pre-translation of an original PLS (from English to Russian) and post-editing from a human translator should be provided to clearly illustrate the proposed approach.
Authors’ answer:
Dear Reviewer,
Thank you for this suggestion. We added a description of the dataset into the Methods subsection “Description of the dataset, selection and randomization of Plain Language Summaries (PLSs)” in yellow highlight. We have also added additional information about Memsource functionalities, a flow chart to illustrate the overall approach more clearly, a graph illustrating the part of the results in the Discussion section, and added screen shots and examples of PLS into the appendix. We hope these changes make the approach clearer.

Reviewer 3 Report
-please add block diagram of the proposed reserach
-please add block diagram of the proposed method
-please add photo of application of the proposed research.
-please add information about future analysis.
-point out what are you done in conclusion section
-please put at least 30 references, 50% of them should be from Web of Science 2018-2021
Author Response
Cover / rebuttal letter to Reviewer 3
Dear editors and reviewers,
With this letter we thank you for the detailed peer-review process and helpful suggestions for the improvement of our manuscript “Assessing human post-editing effort to compare performance of three machine translation engines for English to Russian translation of Cochrane plain language health information: results of a randomised comparison” by Liliya Eugenevna Ziganshina, Ekaterina V. Yudina, Azat I. Gabdrakhmanov, and Juliane Ried.
We are uploading the revised version to the Editorial manager.
Our answers to reviewers’ comments and suggestions:
Reviewer 3
Points 1 and 2:
- “-please add block diagram of the proposed reserach
- -please add block diagram of the proposed method.”
Authors’ answer:
Dear Reviewer,
Thank you for your suggestions.
We added a flow diagram as Figure 1.
Point 3:
- “-please add photo of application of the proposed research.”
Authors’ answer:
Dear Reviewer,
Thank you for this suggestion. We requested permission from Memsource and added relevant screen shots.
Point 4:
- “-please add information about future analysis.”
Authors’ answer:
Dear Reviewer,
Thank you for this comment. We expressed our thinking about future work in the final paragraph of the Discussion section (in yellow highlight).
Point 5:
- “-point out what are you done in conclusion section.”
Authors’ answer:
Dear Reviewer,
Thank you for this comment. We consulted once again with the MDPI Informatics requirements for research articles which indicate:
“Conclusions: This section is not mandatory, but can be added to the manuscript if the discussion is unusually long or complex.”
We opted to add conclusions to provide a concise presentation of our findings, but did not repeat what we did in our study.
Point 6:
- “-please put at least 30 references, 50% of them should be from Web of Science 2018-2021.”
Authors’ answer:
Dear Reviewer,
Thank you for this comment. We consulted again with the MDPI Informatics requirements for research articles and there is no requirement for a minimum number of references. We included new references into the revised version. We have now 26 references, of which 12 are from the period 2018-2020 including, as encouraged by the Journal, and citations of online data accessed 2020-2021.

Reviewer 4 Report
While I found some aspects of this article interesting, it has little content from a computer science or Natural Language Processing point of view. But I acknowledge that the information it contains can be useful to people designing a MT-aided translation process and therefore has merits for publishing.
I understand that the authors are not Machine Translation specialists, but I think they would make their paper more useful by making it follow more closely Natural Language Processing research methods.
In particular, I regret that the authors did not include an open source system in their comparison. The MT systems tested being online commercial products, they can be changed at any point in time without users even being noticed of it. This will make future reproducibility and comparisons difficult. On the other hand, it is true that most open-source MT systems are research systems that are not very easy to use for the non-specialist.
Another issue, is the presentation of results and the chosen measures for quality of translation. There are well established measures for evaluating translation quality:
- Adequacy/Fluency
- HTER (Human-targeted Translation Error Rate)
- Time taken for Post-editing the translations
- BLEU
But the authors use none of them and rather provide rather obscure metrics. One is the Translation QE measure of the Memsource commercial product (whose algorithm is unknown). The other metric is an estimation of the amount of text that had to be post-edited. But how this amount was computed and what it means is rather obscure. The text only mention that e.g “if text fell into the 0%-49% range, it was substantially edited”, which is very vague.
As for the measures above, Adequacy/Fluency require human evaluation and is therefore rather costly. Measuring the "Post-edition time" would have had to be done during the experimentation phase, therefore I suppose it is too late now (unless Memsource keep this information). However, HTER would not be very difficult to compute. It is simply the TER-edit distance between the MT output and the post-edited translation. This TER distance can be computed from tools such as https://github.com/snover/terp
Another issue with the experiment design is that each system is evaluated on differents PLS. I understand that this is to avoid wasting the effort of post-editing one text several times. But it introduces some variance in the evaluation, and I think the authors should discuss this point in the article. In particular I can see from the results table that the DeepL PLS have a higher proportion of TM matches than the ones for MS translate. Could this be that MS translate got "unlucky" and was assigned PLS that were harder to translate than the average? I would like to see some statistics for understanding this variance. In particular:
- Average Length (in words) of the PLS
- Average sentence length
- Average length of the PLS assigned to each MTE
- Average sentence length for each MTE
- Some estimation of the variance of the translation score across PLS (e.g. standard deviation of the HTER scores for each MTE)
As for more minor aspects that could be improved:
- The authors forgot to remove the "0. Template" section at the beginning of the text
- The results would be much easier to read in a graphical form (e.g bar charts)
- I feel the authors sometime assume the reader is familiar with Memsource. They should explain a bit more. e.g is the MTQE metric per line or per PLS? what is the process for TM matching, etc.
Finally, I would like to mention to the authors that the data they are creating can be highly valuable to Machine Translation Researchers. I do not know what is Cochrane's stance on opening this data, but to give some examples from the tasks of WMT2020 (http://www.statmt.org/wmt20/), your post-edited data could be used for:
- Training automatic post-edition system (http://www.statmt.org/wmt20/ape-task.html)
- Evaluating the quality of Translation Metrics (http://www.statmt.org/wmt20/metrics-task.html)
- Training MT systems on medical data (http://www.statmt.org/wmt20/biomedical-translation-task.html)
Therefore, releasing this data as an open ressource could be very valuable (and allow the authors to get further recognition for their work). The authors could consider contacting the organizer of the above tasks if they are interesting in discuss this.
In conclusion, my main request to the author would be to compute the HTER scores for each MTE and discuss the variance introduced by the fact that the MTE are not evaluated on the same PLS.
Reference for HTER:
Matthew Snover, Bonnie Dorr, Richard Schwartz, Linnea Micciulla, and John Makhoul, "A Study of Translation Edit Rate with Targeted Human Annotation," Proceedings of Association for Machine Translation in the Americas, 2006.
Author Response
Cover / rebuttal letter to Reviewer 4
Dear editors and reviewers,
With this letter we thank you for the detailed peer-review process and helpful suggestions for the improvement of our manuscript “Assessing human post-editing effort to compare performance of three machine translation engines for English to Russian translation of Cochrane plain language health information: results of a randomised comparison” by Liliya Eugenevna Ziganshina, Ekaterina V. Yudina, Azat I. Gabdrakhmanov, and Juliane Ried.
We are uploading the revised version to the Editorial manager.
Our answers to reviewers’ comments and suggestions:
Reviewer 4
Dear Reviewer,
Thank you very much for your positive assessment and appreciation of our manuscript and very thorough and helpful suggestions.
Point 1:
- “In particular, I regret that the authors did not include an open source system in their comparison. The MT systems tested being online commercial products, they can be changed at any point in time without users even being noticed of it. This will make future reproducibility and comparisons difficult. On the other hand, it is true that most open-source MT systems are research systems that are not very easy to use for the non-specialist.
Another issue, is the presentation of results and the chosen measures for quality of translation. There are well established measures for evaluating translation quality:
Adequacy/Fluency
HTER (Human-targeted Translation Error Rate)
Time taken for Post-editing the translations
BLEU
But the authors use none of them and rather provide rather obscure metrics. One is the Translation QE measure of the Memsource commercial product (whose algorithm is unknown). The other metric is an estimation of the amount of text that had to be post-edited. But how this amount was computed and what it means is rather obscure. The text only mention that e.g “if text fell into the 0%-49% range, it was substantially edited”, which is very vague.
As for the measures above, Adequacy/Fluency require human evaluation and is therefore rather costly. Measuring the "Post-edition time" would have had to be done during the experimentation phase, therefore I suppose it is too late now (unless Memsource keep this information). However, HTER would not be very difficult to compute. It is simply the TER-edit distance between the MT output and the post-edited translation. This TER distance can be computed from tools such as https://github.com/snover/terp.”
Authors’ answer:
Thank you for these compelling suggestions. We considered them very carefully, and addressed them in the Methods and Results section. We added short descriptions (characteristics) of the three MTEs and a rationale why we chose them into the Methods section, subsection MTE for comparison. We added more detailed descriptions about Memsource features in a subsection Memsource features. We gave a rationale and more specific explanation of our evaluation method in a subsection Principle approach of the study and evaluation method, and in the Results section for each type of analysis. (all highlighted in yellow in the manuscript)
Point 2:
- “Another issue with the experiment design is that each system is evaluated on differents PLS. I understand that this is to avoid wasting the effort of post-editing one text several times. But it introduces some variance in the evaluation, and I think the authors should discuss this point in the article. In particular I can see from the results table that the DeepL PLS have a higher proportion of TM matches than the ones for MS translate. Could this be that MS translate got "unlucky" and was assigned PLS that were harder to translate than the average? I would like to see some statistics for understanding this variance. In particular:
- Average Length (in words) of the PLS
- Average sentence length
- Average length of the PLS assigned to each MTE
- Average sentence length for each MTE
- Some estimation of the variance of the translation score across PLS (e.g. standard deviation of the HTER scores for each MTE)“
Authors’ answer:
Dear Reviewer, thank you very much for these considerations and suggestions. We looked into these suggestions very carefully. We added more information about the heterogeneity of Cochrane PLSs in the Methods section, subsection: Description of the dataset, selection and randomization of Plain Language Summaries (PLSs), and calculated average word count per PLS per MTE as arithmetic means with standard deviation and medians with interquartile range as a measure of dispersion. We showed that the randomised design of the study worked and despite the natural heterogeneity of the Cochrane PLSs each of 30 PLSs randomised to each MTE were statistically homogeneous. We acknowledge that the sentence length can impact MT performance. Unfortunately, we have not kept copies of all pre-translated and edited PLS, only of their analyses, and do not have sentence length data to assess PLS heterogeneity of our dataset from that perspective.
Minor points:
- “The authors forgot to remove the "0. Template" section at the beginning of the text.”
Authors’ answer:
Dear Reviewer,
Thank you for noticing this error on our part, we deleted the lines belonging to the template.
- “The results would be much easier to read in a graphical form (e.g bar charts)
Authors’ answer:
Dear Reviewer,
Thank you for this suggestion. We added Figure 2 to the Discussion section and hope this graph helps read the results.
- “I feel the authors sometime assume the reader is familiar with Memsource. They should explain a bit more. e.g is the MTQE metric per line or per PLS? what is the process for TM matching, etc.”
Authors’ answer:
Dear Reviewer,
Thank you for this comment. We answered this in our answer to your major point 1, please see above.
- “Finally, I would like to mention to the authors that the data they are creating can be highly valuable to Machine Translation Researchers. I do not know what is Cochrane's stance on opening this data, but to give some examples from the tasks of WMT2020 (http://www.statmt.org/wmt20/), your post-edited data could be used for:
- Training automatic post-edition system (http://www.statmt.org/wmt20/ape-task.html)
- Evaluating the quality of Translation Metrics (http://www.statmt.org/wmt20/metrics-task.html)
- Training MT systems on medical data (http://www.statmt.org/wmt20/biomedical-translation-task.html)
Therefore, releasing this data as an open ressource could be very valuable (and allow the authors to get further recognition for their work). The authors could consider contacting the organizer of the above tasks if they are interesting in discuss this.”
Authors’ answer:
Dear Reviewer,
Thank you for this suggestion. Our Cochrane reviews and their translations are unfortunately copyrighted and require a license for use. But the PLSs and scientific abstracts are freely available from our websites and upon request for research purposes. It would not make so much sense for us to make bi-lingual data available at a certain point in time as a package, as it is constantly updated and added to, but we are certainly open to sharing bi-lingual data for research purposes upon request, and have provided data to researchers, including for WMT, in the past.

Round 2
Reviewer 3 Report
-it is good idea to add photo of application
-title of the figure should be below figure (Figure 1)
-Figure 1 can be drawn better for example in Power Point and CTRL+C, CTRL+V to Paint program.
-it is good idea to add steps of processing.
Author Response
Cover / rebuttal letter to Reviewer 3
Dear editors and reviewers,
With this letter we thank you for the detailed peer-review process and helpful suggestions for the improvement of our manuscript “Assessing human post-editing effort to compare performance of three machine translation engines for English to Russian translation of Cochrane plain language health information: results of a randomised comparison” by Liliya Eugenevna Ziganshina, Ekaterina V. Yudina, Azat I. Gabdrakhmanov, and Juliane Ried.
We are uploading the revised version to the Editorial manager – Revision 2 (minor).
Our answers to reviewers’ comments and suggestions:
Reviewer 3
Points 1 and 4:
1.“--it is good idea to add photo of application
- - it is good idea to add steps of processing.”
Authors’ answer:
Dear Reviewer,
Thank you very much for your positive assessment and appreciation of our manuscript and our work on your suggestions within Revision 1.
Points 2 and 3:
2.“-title of the figure should be below figure (Figure 1)
- -Figure 1 can be drawn better for example in Power Point and CTRL+C, CTRL+V to Paint program.”
Authors’ answer:
Dear Reviewer,
Thank you for these suggestions. We moved both figure titles below the figures (Figure 1 and Figure 2, highlighted in yellow in the revised manuscript 2) and redraw the Figure 1. We upload ZIP archive of both figures and hope that redrawn Figure 1 meets the standards. The ZIP archive contains both figures as JPG and PPTX files.
